# Combating Tuberculosis via Restoring the Host Immune Capacity by Targeting *M. tb* Kinases and Phosphatases

**DOI:** 10.3390/ijms252212481

**Published:** 2024-11-20

**Authors:** Shahinda S. R. Alsayed, Hendra Gunosewoyo

**Affiliations:** 1Curtin Medical School, Faculty of Health Sciences, Curtin University, Bentley, Perth, WA 6102, Australia; 2Curtin Health Innovation Research Institute, Faculty of Health Sciences, Curtin University, Bentley, Perth, WA 6102, Australia

**Keywords:** mycobacterial kinases, mycobacterial phosphatases, host immune response, mycobacterial kinase inhibitor, mycobacterial phosphatase inhibitor, PknG, MPtpA, MPtpB, SapM

## Abstract

*Mycobacterium tuberculosis* (*M. tb*) is a remarkably versatile pathogen that possesses a unique ability to counteract the host’s defence mechanisms to control the infection. Several mycobacterial protein kinases and phosphatases were found to play a key role in impeding phagosome maturation in macrophages and accordingly blocking the phagosome–lysosome fusion, therefore allowing the bacteria to survive. During phagocytosis, both *M. tb* and the host’s phagocytic cells develop mechanisms to fight each other, resulting in pathogen elimination or survival. In this respect, *M. tb* uses a phosphorylation-based signal transduction mechanism, whereby it senses extracellular signals from the host and initiates the appropriate adaptation responses. Indeed, the ability of *M. tb* to exist in different states in the host (persistent quiescent state or actively replicating mode) is mainly mediated through protein phosphorylation/dephosphorylation signalling. The *M. tb* regulatory and defensive responses coordinate different aspects of the bacilli’s physiology, for instance, cell wall components, metabolic activity, virulence, and growth. Herein, we will discuss the implication of *M. tb* kinases and phosphatases in hijacking the host immune system, perpetuating the infection. In addition, the role of PknG, MPtpA, MPtpB, and SapM inhibitors in resetting the host immune system will be highlighted.

## 1. Introduction

Signal transduction is a cellular mechanism that is critical for adaptation of bacterial pathogens to extracellular environmental changes [1]. It involves sensing a signal or input via a sensor protein, followed by generating an intracellular response or output via a transducer protein [1]. This process is governed by bacterial protein kinases and phosphatases that mediate protein phosphorylation and dephosphorylation, respectively [2]. In bacteria, the stimulus–response adjusting mechanism is mainly regulated by a two-component system, comprising histidine kinase sensors and their corresponding response regulators (transducers) [2]. In this regard, a histidine kinase is stimulated by a particular intracellular or environmental signal, which results in autophosphorylating a key histidine residue [3]. This phosphorylated histidine then serves as a substrate to the cognate response regulator, wherein an aspartate residue is autophosphorylated. In response to this phosphorylation cascade, the response regulators, which are mainly DNA binding proteins, trigger the expression of certain genes, bringing forth the requisite response [3]. In eukaryotes, the backbone of the signal transduction network consists of serine, threonine, and tyrosine (Ser/Thr/Tyr) protein kinases and their coupled phosphatases [2]. The histidine/aspartate (His/Asp) phosphorylation (two-component system) was previously thought to be exclusive to prokaryotes, whereas the Ser/Thr/Tyr phosphorylation was considered a eukaryotic trait. However, both systems were detected in numerous prokaryotic and eukaryotic cells [3,4,5].

In *M. tb*, the pathogenic success mostly relies on the ability of the mycobacteria to sense and adjust to the complex and dynamic environmental cues of the host [3]. Accordingly, the *M. tb* possesses a wide repertoire of an advanced intracellular signal transduction network encompassing (1) eleven “eukaryotic-like” serine/threonine protein kinases (STPKs); (2) twelve classic prokaryotic signalling machinery (two-component system protein kinases); (3) a single protein tyrosine kinase (PtkA); (4) two protein tyrosine phosphatases (MPtps); and (5) one Ser/Thr phosphatase (PstP) [3,4,5,6]. The presence of Ser/Thr/Tyr protein kinases and a His/Asp two-component system enables the mycobacterium to fine-tune its response, depending on the cellular environment. Indeed, *M. tb* employs the His/Asp phosphorylation system when a rapid short-term response is required, whilst the Ser/Thr/Tyr phosphorylation comes into play when a stable long-term response is mandatory [3]. The *M. tb* exploitation of the two phosphorylation systems allows for its survival and adaption in complex environments. In fact, the synthetic process of mycolic acids (MAs), which are located in the *M. tb* outer membrane and serve as a safeguard to protect the tubercle bacilli against outside threats, was found to be regulated by protein kinases and phosphatases [7]. In the review article published in 2019, we summarised the role of protein phosphorylation/dephosphorylation in modulating the activity of the enzymes involved in MA biosynthetic machinery in *M. tb* [7]. The current review will cover the crucial role of several *M. tb* kinases and phosphatases in overpowering the host immune system, leading to mycobacterial survival. Additionally, the notion of restoring the host immunity capacity to kill the mycobacteria using inhibitors of these kinases and phosphatases will be discussed.

## 2. The Critical Role of Macrophages in Controlling Infection

Macrophages constitute the immune system’s first-line of defence against microbial invaders in the human body and are the connecting link between innate and adaptive immunity [8]. Pathogen invasion sets off a series of macrophages’ signalling cascades, rendering a hostile environment that undermines the microbial pathogenesis. Indeed, macrophages orchestrate a panoply of innate immune events, such as phagocytosis, phagosome–lysosome fusion, and autophagy, to ensure the pathogen clearance from the body [8]. However, in certain cases, macrophages become overpowered by the invading microbes, thereby failing to eliminate them. As a result, these pathogenic intruders manage to form a safe niche within the host, which constitutes a ticking time bomb that can ultimately transform into a full-blown disease. Phagocytosis is an essential part of the macrophage killing machinery, whereby microorganisms are captured, engulfed, and destroyed [9,10]. The phagocytic process can generally be divided into the following four main phases: (1) microbial recognition; (2) phagosome formation and maturation; (3) phagosome–lysosome fusion (phagolysosome formation); and (4) pathogen degradation [9].

Phagocytosis is initiated upon recognition of distinctive molecular patterns associated with pathogens [9,10]. This recognition is accomplished via several specialised receptors on the cell surface of phagocytes, which in turn trigger signalling cascades that prompt phagocytosis. In this respect, after receptor engagement, the plasma membrane walls off the microorganism via surrounding it, followed by membrane sealing, forming a vacuole where the microbe is internalised [9,10] (Figure 1). This phagocytic vesicle (early phagosome), confining the microorganism, subsequently changes its membrane composition and contents in a process called phagosome maturation, through fusion with endosomes and ultimately lysosomes. A “kiss and run” dynamic process between early phagosomes and endocytic vesicles take place, wherein sequential events of fusion and fission lead to the formation of late phagosomes and recycling of endosomes [10,11]. In fact, early phagosomes procure the necessary proteins required for phagosomal maturation when they fuse with the endocytic vesicles (such as early endosomes, late endosomes, and lysosomes), followed by splitting and recycling of the endocytic organelles [12]. Early phagosomes are characterised by the presence of Rab5, which is a small membrane-based guanosine triphosphatase (GTPase). Rab5 regulates the fusion between early phagosomes and endosomes via recruiting early endosome antigen 1 (EEA1) [10]. It also recruits human vacuolar protein sorting 34 (Vps34), which is a class III phosphoinositide 3-kinase (PI3K) that is responsible for the generation of phosphatidylinositol 3-phosphate (PI3*P*). This lipid prompts phagosome maturation via recruiting other effector proteins, such as Rab7, which is a small GTPase, signifying late phagosomes [10]. In fact, Rab5 is considered the key marker of early phagosome, whereas in mature late phagosomes, Rab7 takes over, mediating the fusion between phagosomes and lysosomes and bringing about the dissociation of Rab5 [9,10,13].

When the late phagosomes fuse with lysosomes, they evolve into microbicidal vacuoles called phagolysosomes. Importantly, vacuolar-type adenosine triphosphatase (V-ATPase or V-type H^+^–ATPases) gradually accumulates on the phagosomal membrane upon maturation [10]. Of note, V-ATPases team up with the human vacuolar protein sorting 33B (VPS33B, Figure 1), which is a crucial component of the phagocytosis process (vide infra) [14,15]. V-ATPases use cytosolic ATP as an energy source to translocate protons (H^+^) into the phagosomal lumen, leading to progressive acidification of phagosomes, in which the phagosomal pH gradually drops from 6.5 (early phagosomes) to 4.5 (phagolysosomes) [10,14]. Sturgill-Koszycki and collaborators demonstrated that the lack of acidification in mycobacterial-containing phagosomes could correlate to the exclusion of V-ATPase from the vesicles [16].

A number of sophisticated mechanisms directed towards degrading and eliminating the pathogen take place in phagolysosomes. Indeed, the phagosome–lysosome fusion results in membrane remodelling (alterations in the characteristics of the phagosomal membrane) and the acquisition of more V-ATPases, in addition to lysosomal hydrolases (proteolytic enzymes), such as proteases, cathepsins, lipases, and lysosomes [10,14]. The low luminal pH of the phagolysosomes can be directly toxic to the microorganism, in addition to being optimal for the hydrolytic enzymes to carry out the degradation process of the phagocytosed pathogen [9,10]. Furthermore, other microbicidal elements come into the picture, including scavenger molecules and reactive oxygen species. Collectively, the oxidative and degradative milieus of the phagolysosomes render the interior environment inhospitable to the invading microorganisms, which ultimately leads to pathogen degradation [10]. The protein-based microbial antigens resulting from this process are then presented to the T-lymphocytes, whereupon the adaptive immune system gets activated [9]. Therefore, the adequate maturation of phagosomes into phagolysosomes (innate immunity) is of paramount importance for the degradation of the pathogen and presenting the degraded antigen to the adaptive immunity, which will subsequently result in eliminating the pathogen. In general, this phagocytic process is efficient in removing the invading microorganisms and maintaining homeostasis. However, several microbes, including *M. tb*, have developed tactics to prevent phagosomal ripening and thwart the phagosome–lysosome fusion [10]. Accordingly, these microbes survive in the host by escaping the macrophage killing schemes, perpetuating the infection.

## 3. Role of *M. tb* Kinases and Phosphatases in Warding off Phagosome Maturation and Preventing Phagosome–Lysosome Fusion

When the tubercle bacilli are inhaled and enter the lungs, they are faced with various subpopulations of phagocytic cells, such as dendritic cells and alveolar macrophages [9]. After the bacilli are phagocytosed, *M. tb* blocks the formation of the final antimicrobial organelles (phagolysosome) by hijacking the killing pathways of the macrophages, which precludes the elimination of the bacilli. In fact, *M. tb* turns the table on the host and uses the immune cells to shield itself and form a safe harbour where it can survive. Indeed, *M. tb* has developed several strategies to subvert the immune response and prevent phagosome maturation from proceeding in a normal way, thereby inducing chronic infection, and persisting in a latency mode [9,17]. The ability of *M. tb* to survive in the macrophages via blocking the phagosome–lysosome fusion was first published by Armstrong and Hart nearly 50 years ago [18]. They demonstrated that nearly 70% of phagosomes containing *M. tb* failed to fuse with lysosomes. Since then, numerous virulence factors, produced by *M. tb*, were found to hamper phagosome maturation and arrest the formation of the ultimate killing organelles [14,19]. In particular, the mycobacterial protein kinase G (PknG) was found to be implicated in the prevention of phagosome–lysosome fusion [20]. In addition, three secreted *M. tb* phosphatases, the mycobacterial protein tyrosine phosphatases A and B (MPtpA, MPtpB), in addition to the secreted acid phosphatase M (SapM), co-ordinately interfere with the stages of development of phagosomes, thereby preventing pathogen destruction [21]. Taken together, the *M. tb*-mediated manipulation of host vesicular trafficking processes leads to a malfunction in the process of antigen presentation, causing an ineffective activation of cell populations that contributes to keeping the infection at bay [22].

### 3.1. PknG

Prokaryotes typically use the His/Asp two-component system to regulate their signal transduction; however, when the *M. tb* genome was sequenced, 11 members of the “eukaryotic-like” Ser/Thr kinase family were also discovered, namely PknA, PknB, and PknD–PknL [23]. Among these STPKs, three kinases, PknA, PknB, and PknG, were found to be implicated in the mycobacterial intracellular growth and survival. Both PknA and PknB regulate the mycobacterial cell wall synthesis, cell division-associated morphological changes, and cell growth [23]. On the other hand, to date, PknG is the only *M. tb* STPK that has been reported to modulate the host’s immune system, as a result of which it enhances the *M. tb* survival in macrophages [3]. PknG is released by *M. tb* into the lumen and cytosol of phagosomes and is considered a key player in the prevention of phagosome–lysosome fusion [24]. Therefore, PknG expression is believed to be correlated to the pathogenicity of mycobacteria [20,25]. Since phosphorylation is a key mechanism that is required for regulating vesicular trafficking, it is likely that PknG phosphorylates a host protein that is involved in the phagosome–lysosome fusion, thereby inhibiting this crucial step [19]. However, after being secreted from *M. tb* in macrophages, the mechanism by which PknG is translocated to the cytosol and its exact action on the host’s substrates and vesicular trafficking machinery are yet to be deciphered [19]. Several reports have established the importance of PknG for mycobacteria to avoid lysosomal delivery in both macrophages and dendritic cells [20,25,26]. Inactivation of PknG by chemical inhibition or gene disruption resulted in a rapid lysosomal delivery and mycobacterial degradation in infected macrophages [20]. Similarly, when dendritic cells were infected with a PknG-deleted (Δ*pknG*) strain, the mycobacterial mutants were largely transferred to lysosomes [26]. In fact, in this study, both macrophages and dendritic cells infected with wild-type mycobacteria remained mainly in non-lysosomal phagosomes. Surprisingly, Majlessi et al. showed that the degree of intracellular trafficking of mycobacteria to lysosomes does not impact the extent to which T-cell responses are generated against mycobacterial antigens [26]. They conceded that their findings are in marked contrast with other studies that have established that lysosomal delivery, in both dendritic cells and macrophages, is a prerequisite for the processing and presentation of antigens. Indeed, they reported that the wild-type mycobacteria that resisted lysosomal transfer in macrophages and dendritic cells gave rise to efficient antigen presentation and T-cell responses identical to the PknG-deficient mycobacterial mutants that were instantly shuttled to lysosomes [26]. Interestingly, while PknG was found to be essential for the survival of *M. tb* after the bacterium is phagocytosed inside the macrophage, in in vitro cultures both the PknG-deleted mycobacteria and the wild-type counterpart survived equally [27]. Other reports unravelled more functional facets of PknG, showing that, in macrophages, PknG promotes latency-like conditions [28,29]. In this respect, PknG mediates persistence via modulating cellular metabolism, resulting in efficient metabolic adaptation under stressful environments, such as hypoxia. In other words, PknG functions as a “stress regulator” by combating different stressful conditions experienced by the mycobacteria. In addition, PknG was shown to play a role in abetting drug tolerance [28]. Indeed, PknG was shown to be a key player in the intrinsic resistance of mycobacteria to several antibiotics [30]. Hence, in addition to inhibiting the mycobacterial survival in macrophages, PknG inactivation may also increase the susceptibility of the bacilli to antibiotics [30]. In another study, deletion of the *pknG* gene in *M. tb* resulted in 40–60% less persisters. In addition, the absence of the *pknG* gene led to a 5–15-fold diminished survival of *M. tb* in a chronically infected animal model of TB treated with anti-TB drugs [31]. Interestingly, in a murine model of latent TB, the ability of *M. tb* to revive after antibiotic treatment was extremely diminished in comparison to wild-type and complemented strains [31].

### 3.2. M. tb-Secreted Phosphatases

#### 3.2.1. MPtpA

Among the virulence factors secreted by *M. tb* into the cytoplasm of host macrophages are three phosphatases, denominated MPtpA, MPtpB, and SapM [15]. These phosphatases are vital for altering the host’s signalling pathways and damping down the immune response, leading to optimal bacillary survival within the host and ultimately pathogenesis. Research efforts from different laboratories led to major leaps in the current understanding of the mechanisms by which these phosphatases contribute to evading immune detection of *M. tb* [15]. Like PknG, these *M. tb*-secreted phosphatases are only required for the in vivo growth of the bacteria. Unlike the traditional essential in vitro *M. tb* targets, which have been the main focus of anti-TB drug discovery to date, these phosphatases are dispensable for the in vitro growth of mycobacteria [15]. However, targeting these phosphatases could provide a leeway to circumvent the drug delivery issue correlated with the thick hydrophobic cell wall of *M. tb* [22]. In addition, inhibiting their immune-related modulating activity could lead to resetting the macrophage’s signalling and restoring the innate host’s defence mechanisms.

MPtpA was first identified when the genome of the H37Rv *M. tb* strain was sequenced, revealing its homology to eukaryotic protein tyrosine phosphatases [15]. *M. tb* secretes MPtpA into the cytosol of host’s macrophages, disrupting key elements involved in phagosome maturation [14]. In fact, the *mptpA* gene was found to be overexpressed upon the entry of *M. tb* into host macrophages [32]. The cognate substrate of MPtpA was found to be VPS33B (Figure 1), a key player in the process of phagocytosis, functioning as a regulator of vesicle trafficking and membrane fusion [14,15]. Indeed, MPtpA and VPS33B were found to be colocalised in *M. tb*-infected macrophages [33]. VPS33B is a protein kinase that is ubiquitously expressed in eukaryotic cells and belongs to class C vesicular sorting protein (VPS-C) complex [14,15]. Dephosphorylating VPS33B by MPtpA inactivates the host protein, resulting in the arrest of phagosome–lysosome fusion [33]. In a subsequent study, Wong et al. identified another key MPtpA target that is partnered with VPS33B, namely V-ATPase [34]. Indeed, the lack of phagosome acidification was found to be directly attributed to MPtpA due to its binding to the H subunit of the V-ATPase machinery that drives the low pH of phagosomal lumen. In this regard, they constructed a two-step process model indicating that MPtpA interaction with the V-ATPase machinery is a precondition for the dephosphorylation of VPS33B within the macrophages [34]. They proposed that, first, the *M. tb*-secreted MPtpA binds to V-ATPase, resulting in an initial disruption in membrane fusion, while being placed in close proximity to its catalytic substrate VPS33B. Thereafter, MPtpA dephosphorylates VPS33B, leading to inactivating the entire membrane fusion/phagosome maturation machinery and its downstream effectors [34]. Consequently, the concerted abolition of the activity of V-ATPase and VPS33B by MPtpA prevents V-ATPase trafficking to *M. tb*-infected phagosomes, perpetuating the infection in host macrophages. Indeed, genetic deletion of *mptpA* in *M. tb* (Δ*mptpA*) impaired the bacillary survival within human THP-1 infected macrophages, in which phagosomes harbouring Δ*mptpA* showed enhanced lysosomal fusion, compared to the parental strain [33]. In addition, macrophages transfected with the Δ*mptpA* knockout strain failed to maintain unacidified phagosomes; therefore, the mycobacteria were continually cleared from macrophages [34].

#### 3.2.2. MPtpB

In contrast to MPtpA, *M. tb*’s other secreted protein tyrosine phosphatase, namely MPtpB, does not have a human ortholog, with only 6% similarity with one human PTP [15]. Interestingly, Koul et al. revealed that the *mptpA* gene is present in different mycobacterial species, including *M. tb* complex and *Mycobacterium smegmatis* (*M. smegmatis*), while the *mptpB* gene is exclusively present in members of the *M. tb* complex, suggesting MPtpB plays a unique role in the biological processes restricted to the *M. tb* complex [35]. Like MPtpA, MPtpB is released into the cytoplasm of *M. tb*-infected macrophages [36]. Beresford et al. have demonstrated that MPtpB possesses a unique phosphatase activity, with triple specificity towards phosphoserine/phosphothreonine, phosphotyrosine, and phosphoinositides (PIs) [37]. In particular, manipulating the host phosphoinositide metabolism is a strategy used by pathogenic microbes to promote their colonisation within the infected macrophages [21]. Indeed, the alteration of PI dynamics by *M. tb* affects the intracellular traffic events correlated with phagosome maturation, enabling the long-term survival of bacteria inside the host. In this regard, PI3*P*, which is an early phagosomal membrane tag that is critical for the downstream events of the maturation process, was found to be dephosphorylated by MPtpB (Figure 1) [21]. PI3*P* is an essential membrane-trafficking regulatory lipid that is generated by PI3K on the host membranes of early phagosomes and endosomes and represents a docking site for some proteins associated with the maturation of phagosomes into phagolysosomes [19]. In addition, MPtpB dephosphorylates PI(3,5)*P_2_*, which is another key lipid component, serving as a marker of late phagosomes, and is required for the ensuing phagosome–lysosome fusion [21].

The importance of MPtpB in the intracellular survival of *M. tb* was experimentally proven by Singh et al. when they constructed an *mptpB* mutant strain of *M. tb* [38]. The authors showed that disrupting the *mptpB* gene impaired the ability of *M. tb* to survive in activated macrophages. However, in resting macrophages, both the mutant strain and the wild-type had similar intracellular growth patterns, suggesting the consequential interplay between the host and *M. tb*. In addition, in guinea pigs, infection with the disrupted *mptpB* mutant strain led to a 70-fold inhibition in bacterial burden in the spleens of infected animals, compared to the animals infected with the parental strain [38]. Reintroducing the *mptpB* gene to the mutant strain enabled the resulting complemented strain to establish infection and raise the survival rates in guinea pigs to levels on par with the parental strain [38].

#### 3.2.3. SAPM

SapM is a “eukaryotic-like” acid phosphatase that is secreted by *M. tb* in the host cell cytosol [39]. Recently, SapM was also found to behave as an atypical alkaline phosphatase [21]. Similar to MPtpB, it is believed that SapM functions as a lipid phosphatase, hydrolysing PI3*P* on the phagosomal membrane, thereby preventing PI3*P* accumulation on phagosomes and blocking phagosome–lysosome fusion [21]. However, how SapM is transferred from the phagosomal lumen, where it is secreted, to the phagosome’s cytoplasmic face, where it interacts with PI3*P*, remains a conundrum [14]. While PI3*P* normally regulates the conveyance of phagocytosed consignments to lysosomes, this trafficking event is pre-empted when the accumulation of PI3*P* is halted by *M. tb*. In fact, a PI3*P*-free environment must be maintained by *M. tb* during its occupation in macrophages to accomplish complete phagolysosomal arrest [14]. In this respect, SapM promotes the continuous removal of PI3*P* from phagosomes-containing bacteria, preventing the anchorage of the proteins required for phagosomes to acquire lysosomal components. Indeed, Vergne et al. demonstrated that dephosphorylation of PI3*P* mediated by SapM prevented recruitment of the effector protein EEA1 to the phagosomal membrane, thereby inhibiting the phagosomes–late endosomes fusion [40]. SapM and MPtpB-induced hydrolysis of PI3*P* was also shown to prevent the Rab5 and Rab7 swapping step that is a prerequisite for the transition to late phagosome [21]. In addition to PI3*P*, SapM was found to hydrolyse another PI, namely PI(4,5)*P*_2_, that is involved in the nascent phagosome formation and its subsequent scission from the plasma membrane [21]. Therefore, SapM-mediated blocking of this critical step during the mycobacterial invasion facilitates the uptake and colonisation of *M. tb* within the host. Of note, SapM showed a broad PI activity when tested against several PIs; however, it displayed more specificity towards PI(4,5)*P*_2_ and PI3*P,* which are essential in the early stages of phagocytosis of *M. tb* and phagosome maturation [21]. In addition, Hu and colleagues supported the SapM’s role in the intracellular survival of *M. tb* by showing that it interferes with autophagy [41]. Similar to phagocytosis, autophagy is a highly conserved natural process that is operational in numerous immune cells, especially in macrophages [42]. They both share common features and serve as host defence mechanisms that are required for fighting infections and maintaining proper homeostasis [43]. In this regard, pathogenic microorganisms, such as *M. tb*, produce virulence factors to combat these killing machineries. Indeed, SapM was shown to block autophagosome–lysosome fusion and suppress autophagy by binding to Rab7 [41]. Taken together, SapM seems to possess a pleiotropic role in the pathogenesis of *M. tb*. Deletion of the *sapM* gene in *M. tb* (*M. tb* Δ*sapM*) led to defects in phagosomal maturation arrest and growth inhibition of *M. tb* in human THP-1 macrophages [44]. Indeed, upon disrupting *sapM* in *M. tb*, the resulting strain was severely attenuated, with an impaired ability to grow or cause pathological damage in guinea pig tissues, compared to the parental strain. The importance of SapM in the pathogenesis of *M. tb* was corroborated when the survival of guinea pigs infected with *M. tb* Δ*sapM* was compared to *M. tb*-infected animals. Indeed, the *M. tb*-infected guinea pigs gradually succumbed to death in 4 months, while not even one *M. tb* Δ*sapM*-infected animal died during the whole duration of the study (7 months) [44].

## 4. Restoring the Host Immune Capacity via Inhibiting *M. tb* Kinases and Phosphatases

### 4.1. PknG Inhibitors

The critical role of PknG in promoting the survival of mycobacteria inside macrophages has spurred researchers to search for PknG inhibitors. Unlike most of the currently used anti-TB drugs that interfere directly with *M. tb* growth, inhibiting PknG may recondition the macrophages to carry out their innate microbicidal activity by delivering the bacilli residing in phagosomes to lysosomes [19]. In other words, inactivating PknG will revert the macrophage to a degradative milieu, in which the bacilli are efficiently destroyed and cleared. As a secreted protein, an additional benefit of targeting PknG is that inhibitors are not required to access the highly impermeable mycobacterial cell membrane [19]. Importantly, despite the high homology between the mycobacterial PknG and the eukaryotic Ser/Thr kinases, PknG possesses a unique kinase domain that is distinct from the eukaryotic kinases [45]. AX20017 (**1**, Table 1), a highly selective PknG inhibitor (IC_50_ = 0.39 µM), was found to bind to this unique domain. Indeed, blocking the activity of PknG by this tetrahydrobenzothiophene led to a rapid mycobacterial transfer to lysosomes, followed by killing the intracellular-residing *M. tb* in a dose-dependent manner without impacting the viability of the macrophages [20]. Similar to the Δ*pknG* mutant in mycobacteria, AX20017 exhibited no inhibitory activity against mycobacterial growth in culture (outside host cells) [20]. Although the topology of the kinase domain of PknG is indeed reminiscent of the eukaryotic kinases, AX20017 is harboured in a narrow pocket that is characterised by a unique set of amino acids that are absent in human kinases [45]. This finding in turn explains the high specificity of AX20017 towards PknG and demonstrates that targeting this *M. tb*’s secreted virulence factor can be successfully achieved without compromising the host’s homologous kinases. In a recent study, the implication of PknG in the latency of the mycobacteria was investigated [31]. Indeed, the inhibitory effect of AX20017 on PknG was shown to impact the mycobacterial survival in in vitro models of dormancy, such as nutrient starvation, persisters, and hypoxia. Interestingly, the addition of AX20017 or deletion of the *pknG* gene demonstrated the ability to suppress the formation of persistent/drug-tolerant *M. tb* populations when combined with different antibiotics in vitro and in infected murine macrophages [31].

Sclerotiorin (**2**, Table 1), extracted from marine fungi, showed an IC_50_ of 76.5 µM on PknG [46]. Expectedly, Sclerotiorin failed to inhibit the mycobacterial growth in in vitro cultures. However, *Mycobacterium bovis* (*M. bovis*) Bacille Calmette–Guérin (BCG)-infected macrophages that were treated with 20 µM Sclerotiorin or AX20017 exhibited a 40% and 54% reduction in the bacterial burden in resting and activated macrophages, respectively. Importantly, the macrophages remained viable in the presence of both inhibitors. When sclerotiorin (20 µM or 40 µM) was combined with rifampicin (RIF), the bacterial clearance was slightly enhanced, compared to using rifampicin alone [46].

To identify PknG inhibitors, Kanehiro et al. screened the PknG inhibitory activities of 80 kinase inhibitors [47]. AZD7762 (**3**, Table 1), R406 (**4**, Table 1), and R406-free base (R406f, **5**, Table 1) stood out as potent PknG inhibitors (IC_50_ = 30.3, 7.98, and 16.1 µM, respectively). The three compounds promoted the lysosomal transfer of the *M. bovis* BCG in murine macrophages. They also inhibited the survival of the mycobacteria in infected human macrophages. In addition, R406 and R406f demonstrated bactericidal activities against the mycobacteria in human macrophages, with no cytotoxicity observed [47].

NU-6027 (**6**, Table 1) was recently identified to target PknG from a phenotypic screening of a library of pharmacologically active small molecules, aimed at discovering novel antimycobacterial agents [48]. NU-6027 was previously shown to potently inhibit the activity of various kinases, including cyclin-dependent kinase 1 and 2 (CDK1/2) [62]. Its antitumour activity was subsequently investigated against numerous human tumour cell lines, in which it displayed potent tumour cell growth inhibition [62]. When evaluated against *M. tb* STPKs, NU-6027 inhibited the autophosphorylation activity associated with both PknG and PknD in a dose-dependent manner (at 100 mM and 50 mM) without affecting the kinase activity of the other tested STPKs [48]. Remarkably, NU-6027 potently induced apoptosis of mycobacteria (*M. bovis* BCG) in THP-1 infected macrophages, which was correlated to an upregulation in the expression of proapoptotic genes in the NU-6027-treated macrophages. Of note, no cytotoxicity was observed towards THP-1 cells when treated with NU-6027 at a 25 µM concentration. In addition to macrophages, NU-6027 inhibited the growth of *M. tb* in mouse tissues [48]. Taken together, the preceding findings substantiate the notion that modulating the host/mycobacterial signalling pathways constitutes an attractive approach for the development of novel anti-TB agents.

### 4.2. MPtpA Inhibitors

Although a large number of MPtpA inhibitors have been identified, these inhibitors generally tend to suffer from low selectivity due to the fact that MPtpA shares a 37% similarity with its human ortholog [15]. However, a chalcone derivative (**7**, Table 1) was reported in 2010 as a highly selective MPtpA inhibitor with an IC_50_ value of 50.2 µM [49]. More importantly, this compound reduced *M. tb* survival by 50% and 77% in infected macrophages at 48 h and 96 h post infection, respectively. Compound **7** also demonstrated negligible cytotoxicity against human THP-1 macrophages, killing only 2% of the cells at a 40 µM concentration [49]. In 2017, a highly specific MPtpA inhibitor (L335M34, **8**, Table 1) was reported by Dutta et al., displaying more than 20-fold selectivity over a panel of tested human protein tyrosine phosphatases (PTPs) [50]. Indeed, L335M34 (**8**) showed an IC_50_ value of 160 nM against MPtpA, while no significant activity was observed against all examined human PTPs at concentrations less than 3 µM. Unsurprisingly, L335M34 was bereft of activity in the standard *M. tb* growth inhibition assays, whilst it markedly supressed the bacillary load, at low micromolar concentration, in *M. tb*-infected macrophages (IC_50_ = 1.38 µM) [50].

### 4.3. MPtpB Inhibitors

Numerous selective MPtpB inhibitors were shown to reverse the altered host immune defence reactions and reduce the intracellular growth and survival of *M. tb* in the macrophages [63,64]. In 2009, Beresford et al. reported the isoxazole-derived compound **9** (Table 1) as a potent selective MPtpB inhibitor (IC_50_ = 7 µM) [51]. This compound showed remarkable reduction of the mycobacterial burden of *M. bovis* BCG in macrophages at concentrations of 20, 80, and 160 µM, causing 42%, 64%, and >90% growth attenuation, respectively, in intracellular mycobacteria [51]. A year later, two studies documented three potent and selective MPtpB inhibitors: **10**, **11**, and **12** (IC_50_ = 1.3, 5.6, and 1.26 µM, respectively, Table 1) [36,52]. The benzoindole derivative **10** and piperazinyl-thiophenyl-ethyl-oxalamide derivative **11** demonstrated the ability to overturn the weakened immune response caused by the mycobacterial phosphatases, recapitulating the *mptpB* deletion effects and inhibiting the growth of TB in host cells [52]. Indeed, both compounds **10** and **11** showed nearly total impairment of *M. tb* growth in murine macrophages at a 10 µM concentration, without impacting the viability of macrophages at concentrations up to 100 µM. Predictably, both compounds failed to inhibit the growth of *M. tb* in in vitro cultures at concentrations >100 µM, indicating the unique ability of these compounds to reduce the intercellular *M. tb* survival in macrophages by hindering the MPtpB ability to alter the host immune defences [52]. On the other hand, compound I-A09 (**12**, Table 1) was identified as a potent and specific MPtpB inhibitor (IC_50_ = 1.26 µM) that was able to overcome the perturbation of the host immune surveillance mechanism induced by MPtpB [36]. Interestingly, this compound not only recapitulated the phenotype of the *mptpB*-deleted mutant in active macrophages infected with the *M. tb* Erdman strain, but it also inhibited the bacillary load of *M. tb* in resting macrophages by 90% relative to the untreated cells, while the viability of macrophages remained unaffected. As expected, I-A09 exhibited no activity against *M. tb* in extracellular cultures [minimum inhibitory concentration (MIC) > 100 µM] [36]. In 2013, a hydroxyindole carboxylic acid derivative, **13** (Table 1), was reported to have high selectivity (at least 100-fold) towards MPtpB (IC_50_ = 0.079 µM) over a panel of several protein tyrosine phosphatases (PTPs) [53]. Importantly, compound **13** demonstrated high intracellular efficacy in murine macrophages, restoring the host immune responses challenged by MPtpB. Interestingly, the cellular activity of **13** was found to mimic those of compounds **10**, **11**, and **12**, which are structurally unrelated to compound **13** [53].

Compound **14** (Table 1) was identified by He et al., which displayed an outstanding MPtpB inhibitory potency (IC_50_ = 18 nM) and selectivity, with a more than 10000-fold preference towards MPtpB over a wide panel of 25 phosphatases [54]. This compound also showed excellent activity and specificity in averting the MPtpB function in macrophages. The following year, a benzofuransalicylate derivative, L01Z08 (**15**, Table 1), was reported as a potent and selective MPtpB inhibitor, with an IC_50_ value of 38 nM [50]. As expected, L01Z08 was inactive against *M. tb* in the standard MIC assays, but it significantly diminished the bacterial load in *M. tb*-infected murine macrophages at concentrations <5 µM [50]. In two studies, Tabernero’s group examined the activity compound **16** as a selective MPtpB inhibitor [55,56]. They reported that this compound exhibited dose-dependent efficacy in inhibiting the intracellular *M. bovis* BCG bacterial load in murine macrophages up to 84%, with no activity against extracellular mycobacterial growth, which confirms its exclusive intracellular activity [55]. More importantly, they found that this compound inhibits the bacterial load in human macrophages infected with drug-sensitive (DS) and drug-resistant (DR) *M. tb* by 63% and 74%, respectively. When combined with first line drugs isoniazid (INH, dose = 0.1 µg/mL) and RIF (dose = 0.3 µg/mL), compound **16** (dose = 5 µM) drastically potentiated the inhibition of BCG mycobacterial burden in mouse macrophages from 25% (INH and RIF only) to >93% (INH, RIF, and compound **16**) [55]. Interestingly, compound **16**-treated macrophages demonstrated a prolonged presence of PI3*P*, which is crucial for phagosomal maturation and infection clearance. In animal models, compound **16** as a monotherapy reduced the mycobacterial burden in acute and chronic guinea pig models and showed good oral bioavailability and no adverse drug effects [55]. Apart from *M. tb* complex, compound **16** also reduced the mycobacterial burden in macrophages infected with the nontuberculous mycobacteria (NTM) *Mycobacterium avium* (*M. avium*) [56]. It also demonstrated additive effects when combined with RIF or bedaquiline (BDQ), inhibiting the intracellular mycobacterial burden of both *M. avium* and *M. tb* by 50%, compared to monotherapy with antibiotics [56].

Two groups reported two natural products, Kuwanon G (**17**) and Fusarielin M (**18**, Table 1), as potent inhibitors against MPtpB (IC_50_ = 0.83, and 1.05 µM, respectively) [57,58]. Interestingly, Kuwanon G inhibited the growth of *M. tb* in vitro at an MIC value of 32 µg/mL and exhibited cytotoxicity towards human macrophages at a concentration similar to its MIC value [57]. Therefore, the authors used a non-toxic 10 µg/mL concentration of **17** to assess the *M. tb* survival in macrophages, in which Kuwanon G demonstrated a 61.3% reduction in *M. tb* burden, compared the untreated control [57]. On the other hand, Fusarielin M was shown to selectively inhibit MPtpB and significantly inhibit the intracellular *M. bovis* BCG growth in murine macrophages, reducing the mycobacterial load by 62% at a 20 µM concentration [58].

Recently, Ruddraraju et al. reported the *N*-aryl oxamic acid analogue **19** (Table 1) as a highly potent MPtpB inhibitor (IC_50_ = 0.0064 µM) with >4500-fold selectivity over a large panel of mammalian PTPs [59]. This compound was shown to block the activity of MPtpB in murine macrophages. It also showed no cytotoxicity against mouse embryonic fibroblasts at a concentration as high as 25 µM [59]. In 2023, four rhodanine derivatives, **20**–**23** (Table 1), were reported to show potent inhibitory activities against MPtpB (IC_50_ = 0.48, 0.49, 0.64, 0.35 µM, respectively), with acceptable selectivity towards MPtpB and low cytotoxicity against macrophages and Vero cells [60]. All four compounds demonstrated dose-dependent efficacy in inhibiting the intracellular *M. tb* load in murine macrophages. Interestingly, compound **22** stood out as the most potent inhibitor intracellularly, which was ascribed to its dual inhibition of both MPtpA and MPtpB; it showed a very low MIC value of 30 µg/mL against *M. tb* in in vitro cultures [60]. When mouse macrophages were treated with either compound **22** or RIF, the *M. tb* burden was reduced by 85% compared to the negative control. Importantly, a combination of **22** and RIF led to a >95% inhibition in the intracellular bacterial load, which is better than the inhibitory results observed in RIF or **22** alone [60]. Taken together, since both MPtpA and MPtpB are not essential for *M. tb* growth in vitro, the phenotypic screening wheel is shifted towards utilising a combination of ex vivo macrophage infection models and in vitro enzyme inhibition. This unorthodox approach is indeed one of the currently developing trends, aimed at discovering novel anti-TB compounds.

### 4.4. SapM Inhibitors

In 2019, two SapM inhibitors were identified, namely L-ascorbic acid and 2-phospho-L-ascorbic acid (**24**, Table 1, IC_50_ = 241 and 234 µM, respectively) [21]. In fact, the latter compound significantly reduced the bacterial load/survival of the *M. tb* H37Rv strain in infected THP-1 macrophages, while displaying no detrimental effect on the viability of macrophages. In support of the SapM’s function being likely restricted to promoting the intracellular survival of *M. tb*, compound **24** showed no inhibitory effect on the extracellular growth of *M. tb* at 72 h [21]. More recently, the same group reported Tyrphostin 51 (**25**) and Tyrphostin AG183 (**26**, Table 1) as potent SapM inhibitors (IC_50_ = 6.3 and 8.2 µM, respectively) [61]. Treatment of *M. tb*-infected THP-1 human macrophages with both compounds at 1 µM and 40 µM concentrations led to a substantial decrease in the intracellular mycobacterial burden, phenotyping the *sapM* deletion effects. Expectedly, both inhibitors failed to inhibit the growth of *M. tb* in in vitro cultures. Importantly, both compounds displayed no cytotoxicity, with >70% viability in THP-1 macrophages at concentrations up to 40 µM for 72 h [61]. Overall, contrary to the current antibiotics that are focused on inhibiting traditional essential targets in vitro, the intracellular activities of the preceding MPtpA, MPtpB, and SapM inhibitors constitute a proof of concept that reinstating the intrinsic host signalling machinery could be exploited in eradicating the TB infection.

## 5. Conclusions and Future Directions

In this review, we discussed the critical role of *M. tb*-secreted kinases and phosphatases in establishing infection inside the granuloma and their functions within the phagosomal maturation. The recalcitrant mycobacterial subpopulations prevail during latent TB infections and are phenotypically tolerant towards antibiotics, accounting for the long-drawn-out treatment course of TB. One way to target these mycobacterial persisters is to modulate the phosphorylation/dephosphorylation-based immune evasion mechanisms of *M. tb,* which could extricate the macrophages from the *M. tb* counterforces, potentiating the existing host immune defences against the bacteria. In fact, using host directed therapies (HDTs), which allow for the suppression of *M. tb*-induced host manipulation mechanisms, presents several benefits in comparison to traditional antibiotics, including the following: (a) HDTs can act synergistically with different antibiotics and/or shorten the treatment regimen, (b) HDTs are less likely to generate mycobacterial resistance, and (c) HDTs can be efficacious not only against DS mycobacteria, but also DR and latent mycobacteria. While HDTs that improve immune responses and restore host reactions at the site of infections hold a great potential to eradicate tuberculosis, this promising treatment option is still in its infancy stage, as more comprehensive studies are needed to evaluate their safety in terms of cytotoxicity and long-term side effects. Therefore, HDTs should be further investigated as an adjunct treatment to the current anti-TB drug regimens and not as a sole treatment option. In this respect, targeting *M. tb*-secreted kinases and phosphatases using several chemical probes highlighted herein resulted in unrestrained macrophages with restored inherent killing qualities, especially when used in combination with other antibiotics. This underexploited approach represents a new horizon for the development of novel anti-TB agents effective against the non- or slowly replicating persistent mycobacteria. An Anti-TB drug discovery approach of combining ex vivo macrophage infection models as well as in vitro enzyme inhibition proves to be a powerful method to fight non-replicating TB.

## Figures and Tables

**Figure 1 ijms-25-12481-f001:**
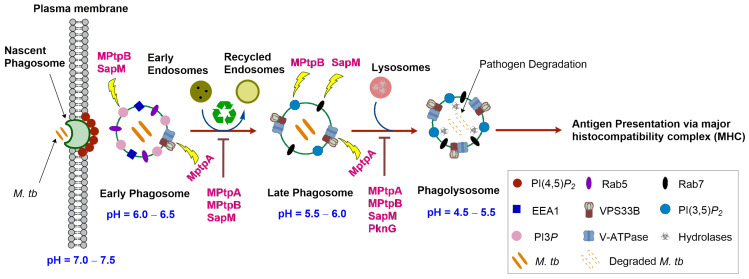
Overview of the phagocytosis of microbial invaders and phagosome maturation, showing *M. tb* secreted kinases and phosphatases interfering with the developmental process of phagosomes. Generally, once engulfed, the pathogen remains confined within early phagosomes, which then undergo a maturation process that involves fusion with endosomes and lysosomes to eventually turn into phagolysosomes, the definitive pathogenicidal vacuoles, followed by fission and recycling of the endocytic vesicles. Early phagosomes are marked by the presence of EEA1, PI3*P*, and Rab5, which contribute to the phagosome–endosome fusion. Upon the procession of phagosomal maturation, V-ATPase accumulate on the phagosomal membrane, lowering the pH of phagosomal lumen. The acidic nature of phagolysosomes constitutes a harsh environment for the microbes and is a prerequisite for the activation of several hydrolytic enzymes. Subsequently, the antigen is degraded and presented, alerting the adaptive immune system. *M. tb*-secreted virulence factors PknG, MPtpA, MPtpB, and SapM impair the phagolysosome fusion, allowing the bacteria to survive. MPtpA impedes phagosome acidification via hydrolysing VPS33B and inhibiting the trafficking of V-ATPase to late phagosomes. MPtpB hydrolyses PI3*P* and PI(3,5)*P*_2_ that mediate the transition to late phagosomes and phagolysosome, respectively. Similar to MPtpB, SapM blocks the preceding transition events via hydrolysing PI(4,5)*P*_2_, PI3*P* and binding to Rab7.

**Table 1 ijms-25-12481-t001:** PknG, MPtpA, MPtpB, and SapM inhibitors involved in restoring the host immune responses.

Inhibitor	Target	Efficacy
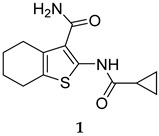 AX20017	PknG IC_50_ = 0.39 µM	- Inhibition of mycobacterial growth in *M. tb*-infected macrophages [20]. - Suppression of *M. tb* persistence when combined with antibiotics in vitro and ex vivo [31].
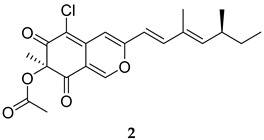 Sclerotiorin	PknG IC_50_ = 76.5 mM	- Reduction in mycobacterial (*M. bovis* BCG) burden in macrophages. - Enhanced mycobacterial clearance when combined with RIF [46].
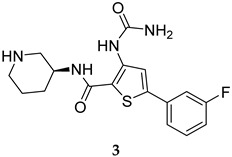 AZD7762	PknG IC_50_ = 30.3 mM	- Enhanced lysosomal transfer and inhibition of mycobacterial (*M. bovis* BCG) survival in macrophages [47].
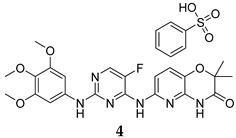 R406	PknG IC_50_ = 7.98 mM	- Enhanced lysosomal transfer and inhibition of mycobacterial (*M. bovis* BCG) survival in macrophages [47].
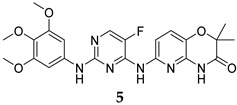 R406-free base	PknG IC_50_ = 16.1 mM	- Enhanced lysosomal transfer and inhibition of mycobacterial (*M. bovis* BCG) survival in macrophages [47].
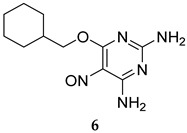 NU-6027	PknG	- Apoptosis of mycobacteria (*M. bovis* BCG) in macrophages. - Inhibition of *M. tb* growth in macrophages and mouse tissues [48].
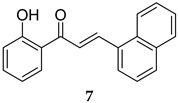	MPtpA IC_50_ = 50.2 mM	- Inhibition of mycobacterial (*M. tb*) survival in macrophages [49].
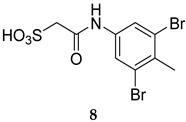 L335M34	MPtpA IC_50_ = 160 nm	- Reduction in bacillary load in *M. tb*-infected macrophages [50].
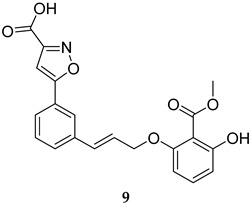	MPtpB IC_50_ = 7 mM	- Inhibition of intracellular mycobacterial (*M. bovis* BCG) growth in macrophages [51].
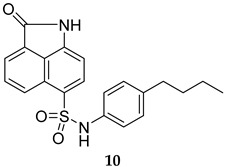	MPtpB IC_50_ = 1.3 mM	- Impairment of *M. tb* growth in macrophages [52].
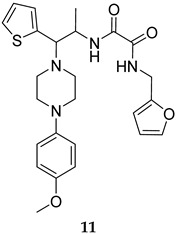	MPtpB IC_50_ = 5.6 mM	- Impairment of *M. tb* growth in macrophages [52].
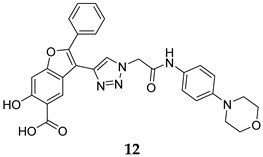 I-A09	MPtpB IC_50_ = 1.26 mM	- Inhibition of mycobacterial (*M. tb*) burden in macrophages [36].
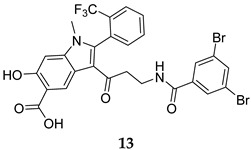	MPtpB IC_50_ = 0.079 mM	- Restoration of host immune responses challenged by MPtpB in macrophages [53].
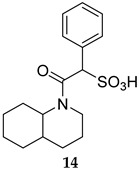	MPtpB IC_50_ = 18 nM	- Reversing the MPtpB function in macrophages [54].
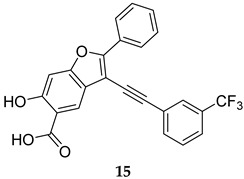 L01Z08	MPtpB IC_50_ = 38 nM	- Inhibition of bacillary load in *M. tb*-infected macrophages [50].
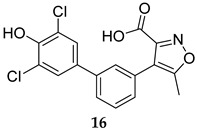	MPtpB	- Reduction in mycobacterial load in macrophages infected with DS and DR *M. tb* in addition to *M. avium*. - Enhanced anti-mycobacterial activity in macrophages when combined with different antibiotics [55,56].
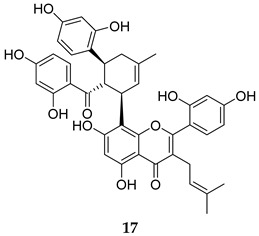 Kuwanon G	MPtpB IC_50_ = 0.83 mM	- Inhibition of *M. tb* burden in macrophages [57].
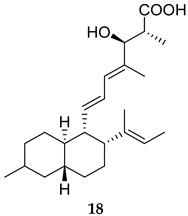 Fusarielin M	MPtpB IC_50_ = 1.05 mM	- Reduction in mycobacterial (*M. bovis* BCG) burden in macrophages [58].
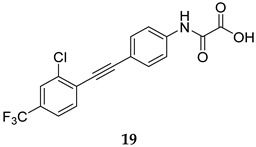	MPtpB IC_50_ = 0.0064 mM	- Blocking the MPtpB activity in murine macrophages [59].
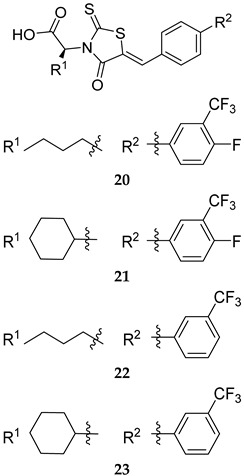	MPtpB 20 IC_50_ = 0.48 mM 21 IC_50_ = 0.49 mM 22 IC_50_ = 0.64 mM 23 IC_50_ = 0.35 mM	- All four compounds showed dose-dependent inhibition of bacillary load in *M. tb*-infected macrophages, with compound 22 exhibiting the highest potency. - When compound 22 was combined with RIF, a further reduction in *M. tb* burden was observed in macrophages [60].
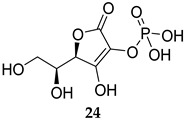 2-phospho-L-ascorbic acid	SapM IC_50_ = 234 mM	- Diminished mycobacterial (*M. tb*) load/survival in infected macrophages [21].
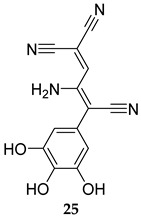 Tyrphostin 51	SapM IC_50_ = 6.3 mM	- Reduction in mycobacterial burden in *M. tb*-infected macrophages [61].
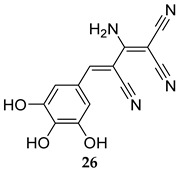 Tyrphostin AG183	SapM IC_50_ = 8.2 mM	- Reduction in mycobacterial burden in *M. tb*-infected macrophages [61].

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
