# Peer review of "Combating Tuberculosis via Restoring the Host Immune Capacity by Targeting M. tb Kinases and Phosphatases"

_ijms, 2024, doi:10.3390/ijms252212481_

Round 1
Reviewer 1 Report
Comments and Suggestions for Authors
This review describes the role of macrophages in controlling infection through the process of phagocytosis and how the pathogen Mycobacterium tuberculosis evades this process through its ability to interfere with the development of the phagosome and prevent phagosome-lysosome fusion. The authors describe the evidence for the role of Mycobacterium tuberculosis secreted kinases and phosphatases in interfering with vesicle trafficking, phagolysosome fusion and phagosome acidification allowing the pathogen to survive within the host cell. Further, compounds that inhibit these kinases and phosphatases have been shown to reduce pathogen survival and intracellular pathogen load.
The overall point of the article is that inhibitors of the Mycobacterium tuberculosis secreted kinases and phosphatases represent a therapeutic target in treatment of tuberculosis.
The article reads well, is well-organized, and the arguments are clear. No criticisms.
Author Response
This review describes the role of macrophages in controlling infection through the process of phagocytosis and how the pathogen Mycobacterium tuberculosis evades this process through its ability to interfere with the development of the phagosome and prevent phagosome-lysosome fusion. The authors describe the evidence for the role of Mycobacterium tuberculosis secreted kinases and phosphatases in interfering with vesicle trafficking, phagolysosome fusion and phagosome acidification allowing the pathogen to survive within the host cell. Further, compounds that inhibit these kinases and phosphatases have been shown to reduce pathogen survival and intracellular pathogen load.
The overall point of the article is that inhibitors of the Mycobacterium tuberculosis secreted kinases and phosphatases represent a therapeutic target in treatment of tuberculosis.
The article reads well, is well-organized, and the arguments are clear. No criticisms.
Response:
We thank the reviewer for their comments.
Reviewer 2 Report
Comments and Suggestions for Authors
The article by Alsayed et al., titled “Combating Tuberculosis via Restoring the Host Immune Capacity by Targeting M.tb Kinases and Phosphatases,” provides a comprehensive overview of the role of mycobacterial protein kinases and phosphatases in manipulating phagosome maturation to enable bacterial survival within macrophages. It explores how M.tb disrupts host immune responses through phosphorylation-based signaling, with a focus on specific kinases and phosphatases such as PknG, MPtpA, MPtpB, and SapM, along with the potential of their inhibitors in restoring host immune function. This article is insightful, though the following suggestions could add significant value:
1. The authors have cited limited sources on the two-component system and should expand this section. Suggested references include “Two-Component Regulatory Systems of Mycobacteria” by Tanya Parish and “An Essential Two-Component Signal Transduction System in Mycobacterium tuberculosis” by Zahrt et al. and may be more.
2. Lines 59-62 are unclear, particularly the phrase: “In the review article published in 2019, we summarised the role of protein phosphorylation/dephosphorylation in modulating the activity of the enzymes involved in the mycolic acids (MAs) biosynthetic machinery in M.tb.” This sentence should be revised for clarity to improve readability.
3. A table listing inhibitors, their selective targets, and their effects on Mycobacterium tuberculosis or relevance to host-directed therapies (HDT) would enhance the article. Specific references should be cited for each inhibitor listed.
4. It would be beneficial to include a section on “Challenges and Future Directions in HDT Development for Mycobacterial Infections Targeting Kinases or Phosphatases,” to provide insights into the barriers and opportunities in advancing HDTs in this field.
Author Response
The article by Alsayed et al., titled “Combating Tuberculosis via Restoring the Host Immune Capacity by Targeting M.tb Kinases and Phosphatases,” provides a comprehensive overview of the role of mycobacterial protein kinases and phosphatases in manipulating phagosome maturation to enable bacterial survival within macrophages. It explores how M.tb disrupts host immune responses through phosphorylation-based signaling, with a focus on specific kinases and phosphatases such as PknG, MPtpA, MPtpB, and SapM, along with the potential of their inhibitors in restoring host immune function. This article is insightful, though the following suggestions could add significant value:
- The authors have cited limited sources on the two-component system and should expand this section. Suggested references include “Two-Component Regulatory Systems of Mycobacteria”by Tanya Parish and “An Essential Two-Component Signal Transduction System in Mycobacterium tuberculosis” by Zahrt et al. and may be more.
Response:
We thank the reviewer for their comment. Both references are now incorporated in the manuscript.
- Lines 59-62 are unclear, particularly the phrase: “In the review article published in 2019, we summarised the role of protein phosphorylation/dephosphorylation in modulating the activity of the enzymes involved in the mycolic acids (MAs) biosynthetic machinery in M.tb.”This sentence should be revised for clarity to improve readability.
Response:
We thank the reviewer for their comment. The following sentence “In fact, the synthetic process of mycolic acids (MAs), which are located in the M.tb outer membrane and serve as a safeguard to protect the tubercle bacilli against outside threats, was found to be regulated by protein kinases and phosphatases.” was added before the highlighted sentence to improve readability and flow.
- A table listing inhibitors, their selective targets, and their effects on Mycobacterium tuberculosisor relevance to host-directed therapies (HDT) would enhance the article. Specific references should be cited for each inhibitor listed.
Response:
We thank the reviewer for their comment. The requested table is now included in the manuscript instead of Figure 2.
- It would be beneficial to include a section on “Challenges and Future Directions in HDT Development for Mycobacterial Infections Targeting Kinases or Phosphatases,” to provide insights into the barriers and opportunities in advancing HDTs in this field.
Response:
We thank the reviewer for their comment. The following section has now been added to the conclusion and future directions section.
“In fact, using host directed therapies (HDT) which allow for the suppression of M. tb-induced host manipulation mechanisms present several benefits in comparison to traditional antibiotics, including: a) HDT can act synergistically with different antibiotics and/or shorten the treatment regimen, b) HDT are less likely to generate mycobacterial resistance, and c) HDT can be efficacious not only against DS mycobacteria, but also DR and latent mycobacteria. While HDT that improve immune responses and restore host reactions at the site of infections hold a great potential to eradicate tuberculosis, this promising treatment option is still in its infancy stage as more comprehensive studies are needed to evaluate their safety in terms of cytotoxicity and long-term side effects. Therefore, HDT should be further investigated as an adjunct treatment to the current anti-TB drug regimens not as a sole treatment option.”
Reviewer 3 Report
Comments and Suggestions for Authors
This paper by Alsayed and Gunosewoyo is a review of the role of M. tuberculosis kinases and phosphatases I the pathogenesis of tuberculosis and the attempts to create anti-tuberculous drugs that target these enzymes. They point out that these bacterial enzymes are not essential for growth om media, are expressed when the bacteria are inside macrophages, and mutants grow normally in media but not in macrophage cell lines. The first 4 pages of the manuscript and figure 1 are a very clear and beautifully written description of the maturation of phagosomes and the role of cellular phosphatases and kinases in this process. They then point out that the bacteria have multiple phosphatases that are expressed when bacteria are intracellular, but they are structurally different from the host enzymes so potentially amenable to drugs. The rest of the paper is devoted to the genetic evidence that these bacterial enzymes are necessary to different extents for TB to survive inside macrophages-like cells, and in some cases for virulence in experimental animals/worms. They then go on to discuss the invitro and in vivo activities of several phosphatase inhibitors including showing the structures in figure 2.
The authors are clearly enthusiastic about this approach and may overstate some of the results. The different drugs were tested in different cell lines, and animal models, and the challenge organisms varied and at least in one case was BCG, but they referred to it as if it was a virulent strain of TB (it is a vaccine). In some of the cites papers the difference in CFU in mutant or wild-type TB inside treated macrophages is modest and of uncertain biological significance. SapM appears to me the most promising target phosphatase as the mutant is actually eradicated by guinea pigs.
I think the authors should include something about how IFNg affects phagosome/lysosome maturation and how this might affect the activity of the drugs they reviewed (if that is known).
One side issue is whether there are mutant bacteria that are resistant to inhibitors. Mutation is the major driver of antibiotic resistance in TB, and it will be very hard to monitor isolates for resistance to this class of drugs as they are not essential genes.
Author Response
This paper by Alsayed and Gunosewoyo is a review of the role of M. tuberculosis kinases and phosphatases I the pathogenesis of tuberculosis and the attempts to create anti-tuberculous drugs that target these enzymes. They point out that these bacterial enzymes are not essential for growth om media, are expressed when the bacteria are inside macrophages, and mutants grow normally in media but not in macrophage cell lines. The first 4 pages of the manuscript and figure 1 are a very clear and beautifully written description of the maturation of phagosomes and the role of cellular phosphatases and kinases in this process. They then point out that the bacteria have multiple phosphatases that are expressed when bacteria are intracellular, but they are structurally different from the host enzymes so potentially amenable to drugs. The rest of the paper is devoted to the genetic evidence that these bacterial enzymes are necessary to different extents for TB to survive inside macrophages-like cells, and in some cases for virulence in experimental animals/worms. They then go on to discuss the invitro and in vivo activities of several phosphatase inhibitors including showing the structures in figure 2.
Response:
We thank the reviewer for their comments.
The authors are clearly enthusiastic about this approach and may overstate some of the results. The different drugs were tested in different cell lines, and animal models, and the challenge organisms varied and at least in one case was BCG, but they referred to it as if it was a virulent strain of TB (it is a vaccine). In some of the cites papers the difference in CFU in mutant or wild-type TB inside treated macrophages is modest and of uncertain biological significance. SapM appears to me the most promising target phosphatase as the mutant is actually eradicated by guinea pigs.
Response:
We thank the reviewer for their comment. We concede that targeting PknG, MPtpA, MPtpB, and SapM as a potential new strategy to tackle tuberculosis is still in its very early stages and further research is needed to prove the feasibility of employing HDT as an adjunct therapy to enhance the TB’s treatment outcomes. The included Table now lists the actual in vitro assays used in the original manuscripts.
I think the authors should include something about how IFNg affects phagosome/lysosome maturation and how this might affect the activity of the drugs they reviewed (if that is known).
Response:
We thank the reviewer for their comment. The topic of how IFNg affects phagosome/lysosome maturation has been reviewed in detail elsewhere in
https://rupress.org/jem/article-abstract/194/8/1123/117/Reprogramming-of-the-Macrophage-Transcriptome-in?redirectedFrom=fulltext
and
https://www.sciencedirect.com/science/article/pii/S1074761308005487.
Therefore, this topic was not included within the scope of the current manuscript.
One side issue is whether there are mutant bacteria that are resistant to inhibitors. Mutation is the major driver of antibiotic resistance in TB, and it will be very hard to monitor isolates for resistance to this class of drugs as they are not essential genes.
Response:
We thank the reviewer for their comment. We agree that mutation in these non-essential genes will be difficult to monitor. There needs to be further research connecting host immune system's ability to contain mycobacteria, especially in the context of latent tuberculosis.